# Immunopathological Alterations after Blast Injury and Hemorrhage in a Swine Model of Prolonged Damage Control Resuscitation

**DOI:** 10.3390/ijms24087494

**Published:** 2023-04-19

**Authors:** Milomir O. Simovic, Zhangsheng Yang, Bryan S. Jordan, Tamara L. Fraker, Tomas S. Cancio, Michael L. Lucas, Leopoldo C. Cancio, Yansong Li

**Affiliations:** 1US Army Institute of Surgical Research, Fort Sam Houston, San Antonio, TX 78234, USA; 2The Geneva Foundation, Tacoma, WA 98402, USA

**Keywords:** polytrauma, hemorrhagic shock, blast injury, immunopathology, organ damage, fluid resuscitation

## Abstract

Trauma-related hemorrhagic shock (HS) remains a leading cause of death among military and civilian trauma patients. We have previously shown that administration of complement and HMGB1 inhibitors attenuate morbidity and mortality 24 h after injury in a rat model of blast injury (BI) and HS. To further validate these results, this study aimed to develop a swine model and evaluate BI+HS-induced pathophysiology. Anesthetized Yucatan minipigs underwent combined BI and volume-controlled hemorrhage. After 30 min of shock, animals received an intravenous bolus of PlasmaLyte A and a continuous PlasmaLyte A infusion. The survival rate was 80% (4/5), and the non-survivor expired 72 min post-BI. Circulating organ-functional biomarkers, inflammatory biomarkers, histopathological evaluation, and CT scans indicated evidence of multiple-organ damage, systemic innate immunological activation, and local tissue inflammation in the injured animals. Interestingly, a rapid and dramatic increase in plasma levels of HMGB1 and C3a and markedly early myocarditis and encephalitis were associated with early death post-BI+HS. This study suggests that this model reflects the immunopathological alterations of polytrauma in humans during shock and prolonged damage control resuscitation. This experimental protocol could be helpful in the assessment of immunological damage control resuscitation approaches during the prolonged care of warfighters.

## 1. Introduction

Trauma-related hemorrhage (TH) is the leading cause of potentially preventable deaths among military and civilian trauma patients during the prehospital phase of care. Blast injury (BI) was the predominant wounding mechanism during recent conflicts, accounting for 70–80% of military casualties in Iraq and Afghanistan [1,2]. The pathophysiology of blast-induced injury is distinctive and appears more complex than most other forms of trauma [3]. Approximately 90% of battlefield casualties die before hospital arrival [4]. Prehospital intervention (<1 h) is critical to reducing overall mortality in TH patients. Despite recent advances in hemorrhage (H) control, TH remains the leading cause of mortality among military and civilian trauma patients [5,6]. However, the lack of effective therapies to create a pro-survival and organ-protective phenotype after severe TH during point-of-injury care is a serious unmet need in military casualties and civilian TH patients.

TH-related tissue injury and ischemia, and initial TH-associated therapeutic interventions (e.g., blood transfusion, volume resuscitation, surgical damage control, instrumentation), instantly and massively activate innate immunity, such as damage-associated molecular patterns [DAMPs; e.g., high mobility group box 1 protein (HMGB1)] and complement cascade (ComC), which represents a crucial driver of inflammation-mediated multiorgan dysfunction syndrome (MODS) and mortality [6,7,8,9]. Growing evidence from our studies and those of others illustrates that early excessive release of HMGB1 and activation of ComC represent a key mechanism regulating the development of inflammation-mediated MODS in civilian and military trauma patients [10,11,12,13,14,15,16,17,18,19,20], rat models of BI [14,21,22,23,24,25,26,27,28], BI+HS [18,19], and swine models of HS [21,25,29,30,31,32]. Recently, we have further shown that early administration of nomacopan, a complement inhibitor [19] and CX-01, an HMGB1 inhibitor [18], attenuate morbidity and mortality 24 h after injury in a rat model of blast injury (BI) and HS.

We also provided preliminary evidence for a potential endotype within trauma patients defined by the triad of complementopathy, endotheliopathy, and coagulopathy, which may serve as a distinguishing prognostic and diagnostic indicator for multiple-organ failure (MOF)/death and a potential therapeutic target for clinical trauma patients [10]. A recent report on closed-head injury caused by dynamic acceleration combined with two levels of HS in mature Yucatan pigs showed diffuse axonal injury, blood-brain barrier breach, and systemic and local inflammation in the brain tissue [33].

While there is an ongoing discussion about optimal resuscitation of the combined TBI and HS [33], some specific working groups provided treatment recommendations for trauma patients in combat settings [34,35]. Trauma, including TBI+HS in experimental settings, determined by survivability/mortality as its exact, not approximated outcome, implies the injury severity. It is how trauma management considerations start. The primary objective of our study was to model a clinically relevant porcine BI+HS trauma with basic resuscitation to further evaluate the efficacy of complement inhibitors and HMGB1 inhibitors in the prolonged care setting (24 h). We intended to follow the “Guidelines for using animal models in BI research [36] and introduce the endotype concept to address the body’s response to polytrauma. We hypothesized that combined blast-induced polytrauma, including BI and controlled H (45% estimated total blood volume), would cause local and systemic inflammation and multiorgan damage. Measurements of hemodynamics, blood chemistry, organ function, inflammatory biomarkers, CT scan, and histological examination were employed to dissect the immunological and pathophysiological changes after BI+HS quantitatively.

## 2. Results

### 2.1. Blast Wave Parameters, Hemodynamic and Chemistry Changes after BI+HS

The shock tube produced consistent open-field blast parameters (Table 1). The mean arterial pressure (MAP) was significantly decreased at 1 and 3 h after BI and hemorrhage, and shock index (SI) was significantly increased from 1 h to 6 h post-injury when compared to baseline (BL) (Table 2). A bolus of Plasma-Lyte A 1 h after BI, followed by continuing fluid resuscitation (1 mL/kg/min), helped MAP recovery. The shock index normalized at 12 h after BI. The enrolled pigs showed variability in the hemodynamic and metabolic response to the combined BI with bleeding. There were no statistically significant differences between lactate and base excess (BE) levels during the entire observation period, although 1–3 h post-injury, the BE levels were decreased, and between 6 and 12 h post-injury, those levels were elevated. The glucose levels were significantly increased at 3 and 6 h after BI (Table 2).

### 2.2. Effect of BI+HS on Circulating Complete Blood Count

The white blood cells (WBCs) showed a trend of increase after the shock phase to 6 h after BI (Figure 1A). The ratio of neutrophils (Figure 1B) to lymphocytes (Figure 1C) between 3 and 6 h after BI was approximately 2.3. The percentage/number of monocytes changed in a pattern similar to lymphocytes (Figure 1D). A decrease in platelets number was observed in the first 3 h after BI (Figure 1E).

### 2.3. Effect of BI+HS on Circulating Complement Activation and HMGB1 Release

A significant increase in C3a in the blood serum at 1 h after BI manifested the activation of complement component 3 (C3) (Figure 2A). A persistent decrease in the activity of the classical complement pathway (CCP, Figure 2B) after BI indicated the activation of CCP. The activity of the lectin complements pathway (LCP, Figure 2D) followed the same pattern as CCP, unlike the alternative complement pathway (ACP, Figure 2C), which did not appear to respond to this polytrauma model. HMGB1 showed very variable levels but a clear tendency of increased concentration in the plasma 1 h after BI (Figure 2E). Relative increases of C3a and HMGB1 levels went in parallel 1 h after BI.

### 2.4. Effect of BI+HS on Circulating End Organ Damage Markers

As shown in Table 3, the pigs had elevated blood levels of troponin I, myoglobin, and aspartate transaminase (AST) starting at 1 h and reaching a significant increase at 6 and 12 h post-BI. Circulating total bilirubin levels were increased, beginning at 1 h and reaching a significant difference at 24 h post-BI (Table 3). Blood glial fibrillary acidic protein (GFAP) started to increase at 1 h, peaked at 6 h, and remained at a high level up to 24 h post-BI. Prothrombin time (PT) was significantly prolonged at the end of the shock phase (1 h post-BI) and the end of the study. Changes in phosphorylated neurofilament heavy chain (p-NF-H), creatinine, and activated partial thromboplastin (aPTT) were significant for all time points.

### 2.5. Effect of BI+HS on Acute Lung Injury

Four pigs survived the observation period of 24 h. One animal died 72 min after BI. Histological evaluation revealed BI+HS-induced multiple-organ damage, including lung, brain, heart, jejunum, liver, and kidney. Macroscopic images showed diffused ecchymotic H extending into the parenchyma (white arrow in Figure 3A). Microscopic photos of H & E staining showed that BI+HS-induced pulmonary injury was characterized by septal thickening, inflammatory cell infiltration, alveolar H, and edema (Figure 3B). We also observed thrombosis, vasculitis, perivascular edema, vascular breach, and H. Semi-quantitative scoring of injury severity on histology further validated these observations (Figure 3C). CT scans demonstrated slight-moderate diffuse to consolidating increased opacity to right accessory and caudal lung lobes after BI+HS (yellow dot and yellow arrows, Figure 3(Db).

### 2.6. Effect of BI+HS on Other Organ Histopathological Alterations

The exposure to the blast wave and hemorrhagic shock resulted in pathological changes typical of neuronal apoptosis, neuroinflammation, and neurodegeneration in the cerebral cortex and hippocampus (Figure 4A). BI+HS induced moderate myocardial edema and degeneration (Figure 4B). Mild renal damage, portrayed by tubular border brush loss, hydropic degeneration, and dilated Bowman’s space filled with hyaline materials and congestion, was observed (Figure 4C). BI+HS produced intestinal injury characterized by hematomas in the wall of the ileum (white arrows) and contusions (turquoise arrows) in the wall of the terminal ileum, as well as submucosal hematoma, denuded, and hemorrhaged villi with lamina propria exudate (Figure 4D). H & E staining revealed disruption of lobular architecture with sinusoidal dilations filled with erythrocytes and Kupffer cells, hepatocytic degeneration and apoptosis, and an area of necrosis and accumulation of inflammatory cells (Figure 4E). Semi-quantitative scoring of injury severity on histology further confirmed these observations (Figure 4F).

### 2.7. Blood Levels of C3a and HMGB1, and Myocarditis and Encephalitis in Non-Survival Animal

Skyrocketing high plasma concentrations of HMGB1 (Figure 5A) and C3a (Figure 5B) in the non-survivor were noted compared to the survivors. Histopathological heart alterations in the pig that died early were distinctly different than in the survivors despite undergoing a similar BI+HS [survivors vs. non-survivor: 235.8 ± 31.7 vs. 211.2 kPa-ms in blast overpressure (BOP) impulse; 45% vs. 45% of estimated total blood volume (ETBV) hemorrhage]. The non-survival animal suffered early severe myocarditis illustrated by interstitial inflammatory cell infiltration with a predominance of macrophages and neutrophils (Figure 5D) and endothelitis and vasculitis with predominant inflammatory macrophage and neutrophil cuffing (Figure 5D), and Purkinje fiber cytotoxic edema and inflammation (Figure 5E). The non-survivor had a lower MAP (22 mmHg) at 1 h post-BI than the survivors (63.5 ± 9.5 mmHg). BI+HS also induced early severe encephalitis depicted by neuroinflammation with a microglial predominance (yellow arrow, Figure 5G), neuronal death (green arrow, Figure 5G), neurodegeneration, neuronal loss, and cerebral cytotoxic edema (orange arrow, Figure H) in the non-survival animal (Figure 5G,H). We noted parallelism between the blood plasma HMGB1 (Figure 5A) and C3a (Figure 5B) levels with the prevalence of myocarditis (Figure 5D,E) and encephalitis (Figure 5G,H). Semi-quantitative scoring of injury severity on histology further affirmed these alterations (non-survivor vs. survivors: 2.8 vs. 1.0 in heart and 2.8 vs. 2.0 in brain cortex).

## 3. Discussion

Our previous findings demonstrate that anti-ComC or anti-HMGB1 therapy increases survival, improves metabolism and hemodynamics, reduces fluid requirements, modulates systemic and local inflammatory responses, and mitigates MOF in porcine models of H or TH during prehospital care (≤6 h) and in a rat TH model during PFC. This study aimed to develop a swine model of BI+HS to further assess these inhibitors’ efficacy.

Compared with in vitro and small animal studies, the pig is an appropriate medical model in many areas of research in human diseases for the following reasons: their anatomical, physiological, immunological, and similarities to humans in disease progress, also the availability of genomic, transcriptomic, and proteomic tools for analysis of this species [37,38]. Swine size enables repeated blood sampling for comprehensive analysis and the use of human medical supportive equipment. Therefore, swine has increasingly become the preferred large animal model of TH to investigate mechanisms and test the efficacy of immunomodulators [31,39,40].

Our previous studies used a porcine isobaric hemorrhage [29,30,40] and a controlled followed by uncontrolled hemorrhage model combined with traumatic injury [31] to study the pathological role of complement activation. Bleeding is the primary cause of morbidity and mortality in surgery and trauma patients [34,41]. Recognizing that TBI and HS after injury are significant causes of death in civilians and military personnel [42], we opted to investigate the pathophysiological and pathological changes of the combination of blast injury and controlled bleeding. A specificity of our study compared to published reports is to hold within a more clinically relevant modeling frame.

The hemodynamic response of injured pigs in our cohort was variable. The primary effect of hemorrhage is reduced cardiac output (CO), and a significant secondary impact of bleeding is reduced arterial O_2_ delivery to body tissues [43]. A decrease in CO during hemorrhage appears primarily because of reduced venous return. This reduction is shown by lowered values for central venous pressure, right atrial pressure, right atrial volume, pulmonary capillary wedge pressure, and left atrial pressure [44,45]. Hemorrhagic shock models vary in the rate, pattern, and amount/volume of blood removal [46,47]. As we planned to conduct volume-controlled/fixed hemorrhage, variations in the bleeding are expected in various animals. Therefore, starting the shock phase immediately after hemorrhage, as presented in Figure 6, was an estimate. The beginning of shock (stage) depends on the individual animal’s response to inflicted bleeding and is related to cardiovascular decompensation. We assumed that the end of bleeding and the start of shock would occur relatively close to each other (temporal proximity). A possible inconsistency of the beginning of the shock phase might explain the observed variability of the hemodynamic response in our cohort of pigs.

Neutrophil to lymphocyte ratio (NLR) is a hematological parameter, an available index of immune response to various systemic inflammatory and non-inflammatory stimuli. The physiological range of NLR in humans is between 1–2, while the values between 2.3–3 occupy a grey zone [48]. Our study’s “grey zone” of the NLR appeared between 3 and 6 h after BI. The increasing number of neutrophils may reflect the activation of complement component 3 (C3) and the generation of C3a. C3a is a mediator of inflammatory processes, where the C3a-receptor is present in inflammatory cells such as granulocytes [49]. C3a is very useful for detecting complement activation since it is relatively stable and expresses an activation-dependent neoepitope [50].

Early coagulopathy predicts mortality in trauma patients. An initial atypical PT increased the odds of dying by 35%, and an initial irregular PTT increased the odds of dying by 326% [51]. We found PT increased at 1 and 24 h after the blast injury. A report showed fibrinolysis activation is more extensive after blast injury than after gunshot wounds [52]. Systemic physiological functions such as heart rate, respiratory rate, and O2 saturation were reduced at 2 min and retrieved by 20 min after 40–50 psi in a shock tube [53]. Musculoskeletal and blast injuries may change systemic arterial pressure [54]. Non-abdominal blast injury to the thorax induces bradycardia, prolonged hypotension, and apnea. A vagal reflex and nitric oxide, a vasodilator released from the pulmonary circulation, appear responsible for this triad [55]. Blast injuries are polytraumatic, and TBI is a common form of this trauma. Clinical blast injury cannot be reduced simply to overpressure. Neither gas-driven shock tube type can reproduce the multiphase flow of debris and fragments propelled by the shock wave [53].

Traumatic brain injury (TBI) is rarely isolated and is usually associated with hemorrhage (H) and injury of other body parts. The coinciding TBI and hemorrhagic shock (HS) are important causes of morbidity and mortality after trauma [33]. Combined TBI and HS modulate vascular tone affecting cardiovascular compensatory functions [56] and increasing the likelihood of respiratory complications [57] and systemic inflammation [58]. Uncoupling cerebral autoregulation and cerebrovascular reactivity after TBI may exacerbate cerebral blood flow upon secondary insult as H-induced hypotension [59]. A reduced cerebrovascular flow after traumatic injury predisposes the brain tissue to secondary insults, including arterial hypotension [60]. Managing HS after TBI is challenging as treatment regimens for these two conditions may be incompatible [61]. A dysfunctional local and systemic blood flow autoregulation after TBI may amplify the harmful effects associated with resuscitation from HS [62]. Fluid resuscitation under dysfunctional local and systemic blood flow autoregulation can result in cerebral edema formation [63]. Therefore, pathogenetic events associated individually with TBI and HS may require different treatment [5,64]. Brain trauma may have a time-dependent effect on the response to hemorrhage as it includes a more significant impact on reactions to immediate hemorrhage than delayed bleeding [65]. Treatment strategies for TBI and HS often discord. A variety of preclinical models have been created to investigate this challenging issue. They differ in modeling TBI, hemorrhage, and animal species used [33,61,62,65,66,67]. Blast injury is a typical military medical condition. The blast wave after explosions can induce a range of TBI from mild concussion to extensive cerebral edema and diffuse axonal injury [68] and can injure thoracal organs [69] and abdominal organs [70]. Experiments in animals indicate that mild brain injuries occur at blast forces similar to the induction point of pulmonary damage [71]. A report showed that veterans with mild TBIs induced by clear blast waves might have more solid evidence of post-concussive symptoms than combatants with mild TBIs caused by blunt force [72]. Evidence of seizures after blast exposure may indicate brain injury, and the hippocampus is one of the primary epileptogenic structures [73]. In our study, pathological features in hippocampal tissue resemble those associated with an increased risk for seizures in humans. Astrocyte activation detectable in temporal lobe epilepsy [73] was also evident in layers of the dentate gyrus in pigs exposed to explosive blast injury [53].

The phosphorylated forms of the p-NF-H and GFAP are axonal and glial cell injury markers, respectively. Serum levels of pNF-H were significantly higher in children with diffuse axonal injury after TBI on an initial CT scan [74]. Plasma pNF-H elevated parallel with the severity of human spinal cord injury (SCI) and reflected more extensive axonal damage [75]. Serum pNF-H increased in patients with brain tissue damage after TBI and peaked at about two weeks to 1 month after injury, correlating significantly with clinical outcomes [76]. Cerebrospinal fluid levels of pNF-H and GFAP are increased in patients with chronic SCI and neurological decline [77]. Ahadi et al. [78] suggest GFAP and pNF-H, among others, be used for diagnosing SCI and injury severity before spinal computed tomography and interventions. Serum levels of pNF-H were used to assess neuropathology in the open [79] and closed [80] models of TBI in rats seven days and 6 h after injury, respectively. We did not find significant differences in the levels of either GFAP or pNF-H in the follow-up intervals observed. pNF-H present in plasma, serum, and CSF may result from several presumed causes. The quantity and rate of pNF-H released from CNS and peripheral nerve tissue, the dynamics of pNF-H transport from source tissues, metabolism of pNF-H in the particular compartments, conveyance of pNF-H into peripheral blood from the CNS, and relative volumes of CSF and peripheral blood may define pNF-H presence in plasma, serum, and CSF. The observation indicated that setting a piece of the pNF-H signal into clotted material is also possible [81].

Investigation of the lung tissues from autopsy cases of fatal close-range blowouts of chemical explosives helped understand micromorphological changes that correspond with the clinical picture and course of blast-induced lung injury in humans. Alveolar and interstitial edema, venous air embolism, bone marrow embolism, and pulmonary fat embolism were observed. Scanning electron microscopy of blast lungs showed alveolar ruptures and expansion of alveolar spaces compared with control subjects. Small perforations of the alveolar wall in diameter of 0.5 and 9 μm and confined intra-alveolar and perivascular hemorrhages were found [82].

As expected, histopathological evidence revealed multiple-organ damage (e.g., brain, heart, lung, intestine, liver, and kidney) in this porcine study of combined blast and hemorrhagic shock. Interestingly, there were pronounced differences in vital organ (heart and brain) damage between the survivors and non-survivor despite undergoing a similar BI+HS. The pathological alterations of heart and brain were positively associated with blood levels of C3a (survivors vs. non-survivor: 401 vs. 2007 ng/mL) and HMGB1 (survivors vs. non-survivor: 57 vs. 1794 ng/mL) at 1 h post-BI, suggesting that early complement activation and HMGB1 release may represent the mechanism underlying the development of myocarditis and encephalitis in non-survivor. These findings are in agreement with our previous reports that demonstrated: (1) early systemic complement activation and HMGB1 release positively correlated with clinical outcomes in combat casualties [18,19] and civilian trauma patients [10,16,20]; (2) BI+HS triggered systemic and local complement activation and systemic HMGB1 release, induced brain injury, and increased mortality in rats, whereas treatment with nomacopan (a C5 inhibitor) or CX-01 (an HMGB1 inhibitor) alleviated this phenomenon [18,19]; (3) HS + voluven resulted in myocarditis and mortality that correlated to systemic and cardiac terminal complement activation (TCA) and plasma levels of TNF-α in swine (unpublished data); and (4) polytrauma + HS-induced myocarditis, encephalitis (unpublished data), and mortality paralleled with systemic and tissue TCA, metabolic acidosis and hypocalcemia in swine, while C1 inhibitor administration ameliorated this phenomenon [31]. Therefore, stratification of disease-associated phenotypes/endotypes in BI+HS may have value for prognostic, predictive, and personalized medicine. Future studies should address phenotype-/endotype-specific aspects of myocarditis and encephalitis using both endotype-specific animal models and endotypes in human trauma cohorts.

This study has several shortcomings due to logistical issues and the small sample size. Volume-controlled hemorrhage requires refinement to increase the consistency of the body’s response to bleeding. Levels of the complement activation components (e.g., C3d, C4d, sC5b-9) in cerebrospinal fluid (CSF) might reduce some areas of uncertainty. Besides blood plasma, serum and CSF would be worth assessing for neuropathological biomarkers. In addition to systemic levels of complement and HMGB1, local expression and distribution of complement, HMGB1, and cytokines would be evaluated using immunohistochemistry, western blotting, and PCR. Further analysis from our ongoing efficacy study with larger sample size and local biomarker measurements is warranted. A review of the effects of unknown factors on our ELISA is also reasonable.

In the current study, we showed that (1) the shock tube produced consistent open-field blast wave parameters and waveforms; and (2) BI+HS-induced immanopathy and multiorgan damage, which may provide a helpful research platform for evaluating the efficacy of innate immunological damage control resuscitation during prolonged field care.

## 4. Materials and Methods

### 4.1. Animal Study

The research complied with the Animal Welfare Act the implementing Animal Welfare regulations, and the research was conducted in compliance with the Animal Welfare Act, the implementing Animal Welfare regulations, and the principles of the Guide for the Care and Use of Laboratory Animals, National Research Council. The facility’s Institutional Animal Care and Use Committee approved all research conducted in this study (approved code: A-18-022; approved date: 9/16/2018). The facility where this research was conducted is fully accredited by AAALAC International.

#### 4.1.1. Animal Surgical, Injury, PDCR, and ICU Procedures

Yucatan minipigs (5 sexually mature females, 5–6 months old, 16–24 kg) were obtained via Sinclair BioResources (Columbia, MO, USA). The animals were acclimated for at least 7 days and checked for acute respiratory infection, manifested by respiratory distress, coughing, and preexisting lung densities at baseline CT.

Indwelling catheters were placed in the left jugular vein (6.5 fr OD, Arrow Int’l, Reading, PA, USA), right carotid artery (8 fr OD, Arrow Int’l, Reading, PA, USA) for fluid/drug administration, and in the femoral artery (8 fr OD, Arrow Int’l, Reading, PA, USA) for arterial sampling and withdrawing blood at a rapid rate for the controlled H.

On the day of the experiment, after overnight fasting with water ad libitum, anesthesia was induced using Telazol (tiletamine/zolazepam 6 mg/kg) and Glycopyrrolate (0.01 mg/kg), then animals were brought to the surgical plane of anesthesia using isoflurane (1–3%). Analgesia was performed by injecting IM Buprenex SR (0.24 mg/kg). The animals were transported to the CT room for baseline CT and then to the operating room (OR) for surgical line placement. During line placement, the anesthesia was transitioned to total intravenous anesthesia (TIVA) comprised of midazolam HCl and propofol, titrated to effect. Next, the animals were transported to the shock tube, exposed to blast injury, and returned to the OR for hemorrhagic shock induction. After the hemorrhage and 30 min of shock period, they were transported to the ICU for the remainder of 24 h for clinical observation and blood sample collection.

As shown in Figure 6A, anesthetized swine were randomized and subjected to a moderate BI (BOP = 350 kPa; t_+_ = 2.5 ms) [34,35,83]. The pigs were placed in the prone position with the right side toward the BOP front, on a networked holder (Figure 6B). All BOP exposures were conducted using USAISR’s shock tube with an 8-foot expansion cone, a 2-foot driven section, and a 2-foot driver section (Applied Research Associates, Inc., Albuquerque, NM, USA, Figure 6C), and a representative blast overpressure waveform was shown in Figure 6D. H was performed using a computer-controlled peristaltic pump (Masterflex, Cole-Parmer, Vernon-Hills, IL, USA) via the femoral artery. Blood was removed via Tygon tubing (E-Lab [E3603] L/S 16, Cole-Parmer, Vernon Hills, IL, USA) into a 1000-mL container. Tubing was primed with a CPD buffer. ETBV was calculated using the following formula: ETBV = weight in kg × 65 mL/kg. Animals were bled (45% ETBV) at a rate of 100 mL/min in 15 min. There were no hemodynamically unstable (mean blood pressure < 60 mmHg) pigs or those with increased lactate levels (>2.5 mm/L) after surgery [33,84]. In developing our porcine polytrauma model, we used Plasma-Lyte A as a resuscitation fluid according to current Tactical Combat Casualty Care (TCCC) and Prolonged Field Care (PFC) guidelines. Damage Control Resuscitation in Prolonged Field Care [85]. The resuscitation regimen is detailed in Figure 6A: After a 30-min-shock period, the animals received a bolus of warmed Plasma-Lyte A if MAP < 55 mmHg, followed by continuous infusion via femoral vein at a rate of 1 mL/kg/min. The timing and regimen of resuscitation are designed to be consistent with current Tactical Combat Casualty Care and PFC guidelines which recommend maintaining casualties in a permissive hypotensive state [34,35,86].

After BI, animals were placed on conventional volume-cycled ventilation. as described below. The animals were continuously monitored and kept on TIVA for the study. A data acquisition system (IDEA; Integrated Data Exchange and Archival system, San Antonio, TX, USA) was utilized for continuous data recording. The following parameters were collected: ECG, arterial waveforms, end-tidal carbon dioxide waveform, pulmonary artery pressure, and temperature. Thoracic CT scans were performed at baseline, and 24 h with a Toshiba Aquilion CT scanner (Toshiba America Medical Systems Inc., Tustin, CA, USA), and 0.5 cm slice images were obtained without contrast.

#### 4.1.2. Biosampling

Blood samples were collected before the blast (B1, baseline) and at 1, 3, 6, 12, and 24 h later. All animals underwent necropsy, and gross organ findings were recorded. Tissue samples (brain, lung, heart, liver, jejunum, and kidney) were collected and processed to permit histological evaluation.

### 4.2. Assays

#### 4.2.1. Analysis of Plasma C3a and HMGB1

Blood C3a concentrations were assessed as described previously [31]; all reagents, including anti-porcine C3a capture antibody, anti-porcine C3a detection antibody, and porcine C3a standard, were purchased from MBM ScienceBridge GmbH (Göttingen, Germany). Plasma C3a levels were measured using ELISA. Circulating HMGB1 levels were determined by ELISA (cat# ST51011, IBL-International, Baldwin Park, CA, USA) according to the manufacturer’s instructions).

#### 4.2.2. Analysis of Complement Functional Activity

Functions of complement classical, lectin, and alternative pathways were examined using the complement system screening kits (cat#: HIT430, HIT431, and HIT432, HycultBiotech, Plymouth Meeting, PA, USA). As per our previous report [16], the serum activity of three complement pathways was measured by ELISA according to the manufacturer’s instructions (HycultBiotech, Plymouth Meeting, PA, USA).

#### 4.2.3. Complete Blood Count (CBC) and Coagulation Parameter Assessment

CBC and coagulation parameters (PT and PTT) were measured using an ABX Pentra 120 hematology analyzer (ABX Diagnostics, Inc., Irvine, CA, USA) and Start-4 (Diagnostica Stago, Rue des Freres Chausson, France), respectively.

#### 4.2.4. Assessment of End-Organ Damage Markers

Lung function (PFR), TBI biomarkers (GFAP, cat#: NBP151153, Thermo Fisher Scientific, Waltham, MA, USA; and pNF-H, cat#: NS170, MilliporeSigma, Burlington, MA, USA), cardiac function (troponin), kidney function (creatinine), liver function (bilirubin and AST), and myoglobin were measured in blood using i-Stat, ELISA, and Dimension Xpand Plus Integrated Chemistry System (Siemens, Holliston, MA, USA), respectively.

#### 4.2.5. Blood Gas and Chemistry Laboratory Assays

Arterial and mixed venous blood gas analysis was performed at the bedside using an iSTAT 300-G blood analyzer (Abbott Point of Care Inc., Princeton, NJ, USA; VetScan CG4+ and CG8+ cartridges, Abaxis Inc., Union City, CA, USA). The following parameters were measured: pH, pCO_2_, pO_2_, O_2_ saturation, hematocrit, hemoglobin, sodium, potassium, chloride, ionized calcium, glucose, base excess/base deficit (BE/BD), and lactate concentration. Blood chemistry (troponin I, myoglobin, total bilirubin, AST, creatinine) was analyzed using Dimension Xpand Plus Integrated Chemistry System (Siemens, Holliston, MA, USA) by a chemistry laboratory in the USAISR. Baseline laboratory values were discussed with the Attending Veterinarian.

### 4.3. Histopathological Evaluation

Tissues were fixed in 10% formalin and were embedded in paraffin. Coronal sections were then cut and stained with hematoxylin-eosin (H&E) [22,23,29,31,87]. Histological images of entire sections for each porcine tissue were recorded with a 10× objective under a slide scanner (Axio Scan. Z1 v1.0, Zeiss, Germany), andrepresentative images of each group were presented (magnification = 400× for lung, brain, heart, liver, and kidney or 100× for jejunum). The changes were semi-quantitatively scored in 30 randomly selected fields at a 400× or 100× magnification by a pathologist. In this study, a group of subjects (uninjured, n = 3) from a previous study with the same species, strain, sex, and vendor, and similar age and bodyweight served as historical control data for histopathological injury comparison.

For the lung injury score, four parameters (alveolar fibrin edema, alveolar hemorrhage, septal thickening, and intra-alveolar inflammatory cells) were scored on each hematoxylin and eosin (H&E) stained slide based on: (1) severity (0: absent; 1, 2, 3 and 4 for increasingly severe changes); and (2) the extent of injury (0: absent; 1: <25%; 2: 25–50%; 3: 50–75%; 4: >75%). The total injury score for each slide was calculated as the sum of the severity plus the extent of the injury.

For the scoring brain injury score, we undertook the approach previously described. Two parts of the brain tissue were scored, including the frontal cortex and hippocampus. Damage was assessed using 5 distinct morphological parameters: neuronal morphological changes (shrinkage of the cell body, pyknosis of the nucleus, disappearance of the nucleolus, and loss of Nissl substance, with intense eosinophilia of the cytoplasm), neuronal loss, cytotoxic edema, vasogenic edema, and inflammatory cell infiltration in the brain cortex and hippocampus. The changes were scored according to their extent (score 0, 1, 2, 3, and 4 for an extent of 0%, <25%, 25–50%, 50–75%, and 75–100%, respectively) and severity of the injury (score 0 = normal histology, score 1 = slight, 2 = mild, 3 = moderate, and 4 = severe alterations).

For the cardiac injury score, five parameters (edema, degeneration, inflammation, congestion, and subendocardial hemorrhage) were scored on each H&E-stained slide for (1) severity (0: absent; 1, 2, 3, and 4 for more severe changes) and (2) the extent of the injury (0: absent; 1: <25%; 2: 25–50%; 3: 50–75%; 4: >75%). The total injury score for each slide was calculated as the sum of the severity and the extent of the injury.

For the hepatic injury score, four parameters, including vascular congestion, hepatocyte death, degeneration, and inflammation, were considered, and these parameters were assayed for severity (score 0 for no change, score 1, 2, 3, and 4 for more severe changes) and for the extent of injury (0: absent; 1: <25%; 2: 25–50%; 3: 50–75%; 4: >75%). The injury score represents the sum of the extent and severity of the injury.

For the jejunum, each slide was scored according to the following scale: 0, normal villi; 1, villi with tip distortion; 2, villi lacking goblet cells and containing Guggenheim’s spaces; 3, villi with patch disruption of the epithelial cells: 4, villi with exposed but intact lamina propria and epithelial cell sloughing; 5, villi in which the lamina propria was exuding; and 6, hemorrhaged or denuded villi.

For the renal injury score, 0 = normal histology; 1 = slight alteration (loss of brush border, mild hydropic degeneration, mild congestion); 2 = mild (intensive hydropic degeneration, mild vacuolization, and interstitial edema); 3 = moderate (nuclear condensation, intensive vacuolization, modulate interstitial edema); 4 = severe (necrotic/apoptotic cells, denudation/rupture of basement membrane) are scored on each H&E stained slide for (1) severity (0: absent; 1, 2, 3, and 4 for more severe changes) and (2) extent of injury (0: absent; 1: <25%; 2: 25–50%; 3: 50–75%; 4: >75%). The total injury score for each slide was calculated as the sum of the severity and the extent of the injury.

### 4.4. Statistical Analysis

Statistical analyses were performed using GraphPad Prism 9.0 (GraphPad Software, Inc., San Diego, CA, USA) and Excel ver. 14.0. Data were analyzed by the Mann–Whitney U test or unpaired *t*-test with Welch’s correction. The data are presented as mean ± SD and tested for the mean difference between study groups. Statistical significance was determined at the 2-sided *p* < 0.05. All data were included, and none were treated as outliers.

## Figures and Tables

**Figure 1 ijms-24-07494-f001:**
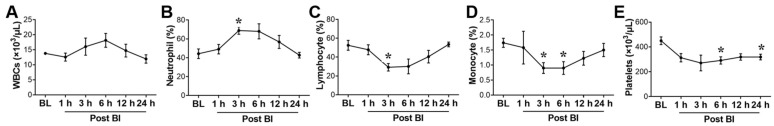
Circulating alterations of WBCs (**A**–**D**) and platelets (**E**) in a porcine BI+HS model. The data are presented as mean ± SD (n = 5). * *p* < 0.05 vs. BL.

**Figure 2 ijms-24-07494-f002:**
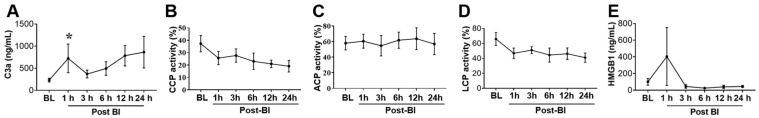
Systemic complement activation (**A**–**D**) and HMGB1 (**E**) in a porcine BI+HS model. CCP, classical complement pathway; ACP, alternative complement pathway; LCP, lectin complement pathway. The data are presented as mean ± SD (n = 5). * *p* < 0.05 vs. BL.

**Figure 3 ijms-24-07494-f003:**
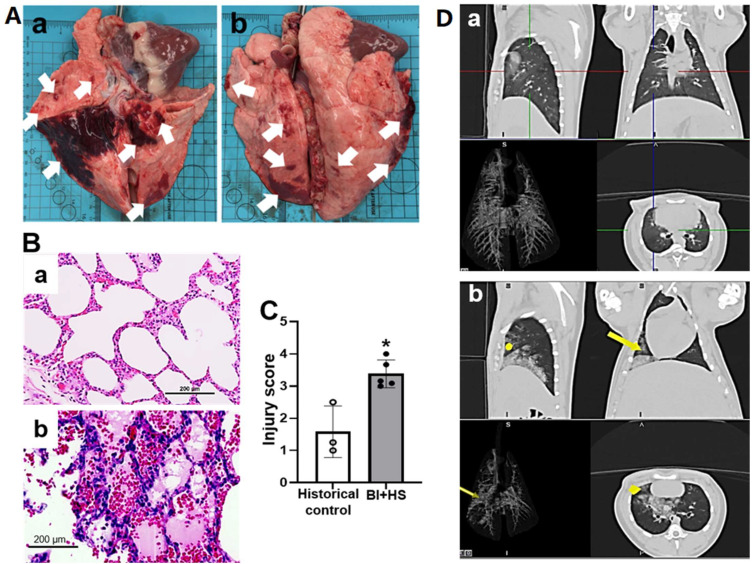
BI+H-induced acute lung injury (a, anterior; b, posterior) in pigs depicted by gross pathological ((**Aa**), anterior; (**Ab**), posterior; white arrows indicating lung injury with diffuse ecchymotic hemorrhage), micropathological alterations of H&E-stained slide [(**Ba**), historical control group (n = 3); (**Bb**), B+HS group (n = 5] and histological injury score (**C**), and CT changes ((**Da**), pre-injury; (**Db**), post-injury; yellow dots and yellow arrows depicting opacity in the right accessory and caudal lung lobes, respectively). The data are expressed as mean ± SD. * *p* < 0.05 vs. historic control. Scale bar = 200 µm.

**Figure 4 ijms-24-07494-f004:**
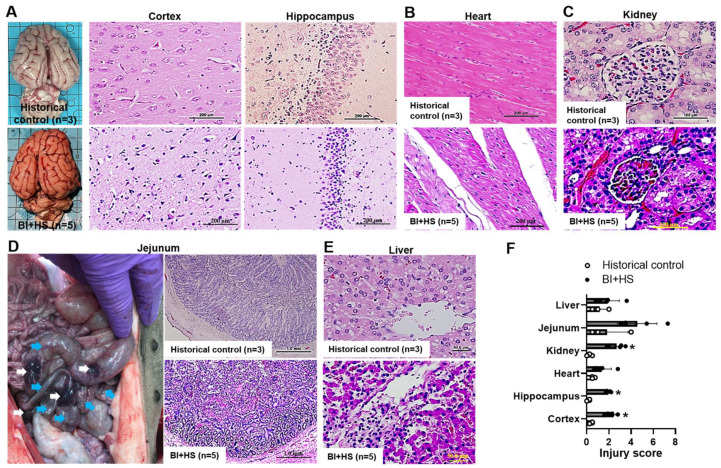
BI+HS induces histological changes in porcine brain ((**A**), scale bar = 200 µm), heart ((**B**), scale bar = 200 µm), kidney ((**C**), scale bar = 100 µm), jejunum ((**D**), scale bar = 1 µm), and liver ((**E**), scale bar = 50 µm). Organ injury scores based on the criteria described in the Materials and Methods (**F**). n = 5. The data are expressed as mean ± SD. * *p* < 0.05 vs. historic control (n = 3).

**Figure 5 ijms-24-07494-f005:**
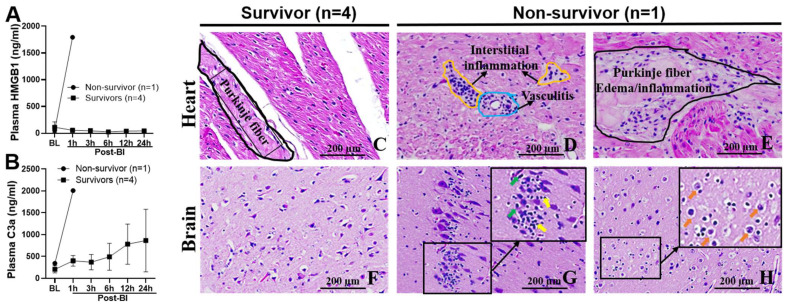
Parallelism of the blood plasma levels of HMGB1 (**A**) and C3a (**B**) without myocarditis (**C**) and neuroinflammation (**F**), and with myocarditis (**D**), Purkinje fiber edema/inflammation (**E**), neuroinflammation ((**G**), yellow arrows)/neuronal death ((**G**), green arrows), and cytotoxic brain edema ((**H**), orange arrows) in a porcine BI+HS model. The data were expressed as mean ± SD.

**Figure 6 ijms-24-07494-f006:**
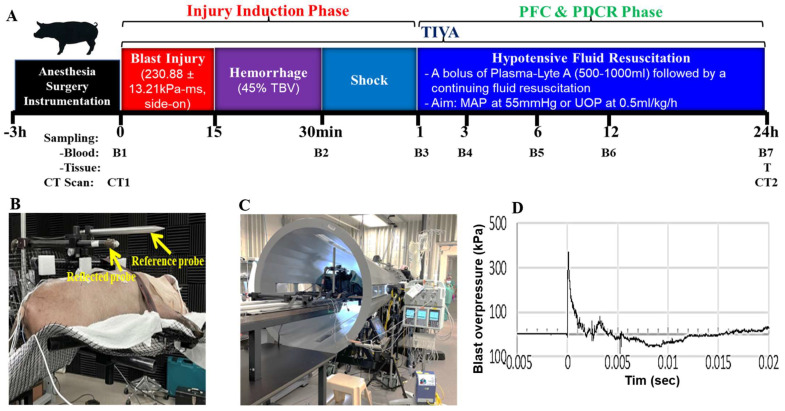
Schematic showing the experimental design and time frame of BI+HS in swine (**A**), pig in mesh holder outside (**B**) and inside (**C**) shock tube, representative blast overpressure wave form (**D**).

**Table 1 ijms-24-07494-t001:** Blast wave parameters obtained from the shock tube.

	Overpressure			Reflected		
	P0 (kPa)	t_+_ (ms)	I (kPa-ms)	P0 (kPa)	t_+_ (ms)	I (kPa-ms)
BI+HS (n = 5)	338.4 ± 22.55	2.09 ± 0.22	230.88 ± 29.55	616.92. ± 77.1	3.55 ± 1.19	472.58 ± 56.64

Notes: BI+HS, blast injury + hemorrhagic shock; I, impulse; P0, peak pressure; t_+_, the positive-pressure phase duration.

**Table 2 ijms-24-07494-t002:** Hemodynamic and blood chemistry parameters in a porcine BI+HS model.

Parameters	Baseline (BL)	1 h Post-BI	3 h Post-BI	6 h Post-BI	12 h Post-BI	24 h Post-BI
MAP (mmHg)	109.4 ± 13.30	55.20 ± 20.29 *	73.25 ± 8.18 *	79.50 ± 20.60	94.25 ± 11.15	87.50 ± 20.34
PP (mmHg)	37.8 ± 7.16	17.6 ± 9.74 *	29.25 ± 4.79	13.75 ± 3.59 *	24.25 ± 9.47	31.5 ± 13.63
Shock Index	1.09 ± 0.12	2.40 ± 0.54 *	1.93 ± 0.34*	1.76 ± 0.31 *	1.43 ± 0.30	1.13 ± 0.22
SpO_2_ (%)	96.40 ± 2.70	86.00 ± 25.88	95.50 ± 2.09	96.75 ± 4.27	96.25 ± 2.50	98.50 ± 1.73
PFR	476.97 ± 82.18	366.36 ± 120.02	477.67 ± 64.00	478.76 ± 26.34	472.00 ± 48.77	507.33 ± 84.68
pH	7.37 ± 0.06	7.36 ± 0.03	7.45 ± 0.06	7.47 ± 0.06	7.49 ± 0.03 *	7.45 ± 0.02
BE/BD	4.80 ± 1.48	3.0 ± 2.65	3.75 ± 3.59	5.25 ± 4.50	6.00 ± 2.45	4.75 ± 3.30
Lactate (mmol/L)	2.13 ± 0.70	4.47 ± 4.21	3.98 ± 3.00	2.35 ± 3.03	0.71 ± 0.19 *	1.08 ± 0.51
K^+^ (mmol/L)	3.96 ± 0.15	4.78 ± 2.04	4.13 ± 0.45	4.03 ± 0.62	3.70 ± 0.82	3.65 ± 0.40
Glucose (mg/dL)	76.00 ± 14.34	83.40 ± 33.89	111.50 ± 17.37 *	122.25 ± 46.65 *	77.75 ± 18.23	63.25 ± 7.32
Hct (%)	26.20 ± 1.48	30.60 ± 1.52 *	25.50 ± 4.20	24.25 ± 3.78	18.50 ± 4.04 *	16.75 ± 2.22 *
Hgb (mmol/L)	8.90 ± 0.49	10.40 ± 0.50 *	8.65 ± 1.42	8.05 ± 1.62	7.50 ± 0.00	4.18 ± 2.25 *

Notes: Data were presented as mean ± SD; n = 4~5. The Kruskal–Wallis test was used to analyze pulse pressure, while the Mann–Whitney test was used to analyze other variables. * *p* < 0.05 vs. BL. BE, base excess/base deficit; Hct, hematocrit; Hgb, hemoglobin; MAP, mean arterial pressure; PFR, PaO_2_/FiO_2_ ratio; post-BI, post-blast injury; PP, pulse pressure.

**Table 3 ijms-24-07494-t003:** Circulating levels of organ injury biomarkers in a porcine BI+HS model.

Biomarkers	Baseline (BL)	1 h Post-BI	3 h Post-BI	6 h Post-BI	12 h Post-BI	24 h Post-BI
Troponin I (pg/L)	17.9 ± 13.59	40.28 ± 30.09	75.67 ± 38.25	374.48 ± 392.49 *	2461 ± 2648.02 *	1711.3 ± 2525.21
Myoglobin (ng/mL)	52.75 ± 9.03	90.00 ± 10.23 *	62.50 ± 10.08	188.75 ± 204.98 *	491.25 ± 333.75 *	905.67 ± 964.93
GFAP (ng/mL)	0.20 ± 0.34	0.59 ± 0.56	0.86 ± 0.99	3.72 ± 5.34	0.86 ± 3.72	0.66 ± 0.74
p-NF-H (ng/mL)	14.54 ± 1.93	14.60 ± 1.56	15.26 ± 0.88	15.20 ± 0.16	15.56 ± 0.74	15.40 ± 0.90
Total bilirubin (mg/dL)	0.09 ± 0.04	0.14 ± 0.08	0.12 ± 0.03	0.10 ± 0.07	0.20 ± 0.08	0.60 ± 0.44 *
AST (U/L)	27.20 ± 22.71	110.00 ± 164.91	35.50 ± 26.34	58.75 ± 6.90 *	91.75 ± 21.69 *	174.00 ± 206.41
Creatinine (mg/dL)	0.57 ± 0.29	0.72 ± 0.40	0.73 ± 0.06	0.79 ± 0.12	0.75 ± 0.15	0.58 ± 0.22
PT (second)	13.38 ± 0.35	14.28 ± 1.88 *	14.20 ± 0.34	13.85 ± 0.31	13.85 ± 0.53	14.48 ± 0.95 *
aPTT (second)	26.00 ± 7.22	25.06 ± 9.43	24.37 ± 4.74	24.58 ± 5.99	27.00 ± 4.38	35.89 ± 13.39

Notes: Data were presented as mean ± SD; statistical analysis was performed by the Mann–Whitney test. * *p* < 0.05 vs. BL. aPTT, activated partial thromboplastin time; AST, aspartate transaminase; GFAP, glial fibrillary acidic protein; p-NF-H, high molecular weight phosphorylated neurofilament; PT, prothrombin time.

## Data Availability

Not applicable.

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
