# Peer review of "Immunopathological Alterations after Blast Injury and Hemorrhage in a Swine Model of Prolonged Damage Control Resuscitation"

_ijms, 2023, doi:10.3390/ijms24087494_

Round 1
Reviewer 1 Report
The authors describe an intersting porcine model of blast injury. However, several aspects must be addressed before publication:
Abstract: Word limit = 200
Introduction: The authors state that: “The primary objective of our study was to model a clinically relevant porcine BI+HS trauma with basic resuscitation in the prolonged care setting (24 hours). We intended to follow the "Guidelines for using animal models in BI research and introduce the endotype concept to address the body's response to polytrauma.” -> However, the authors start the introduction with TBI. Also, a significant proportion of the introduction deals with TBI. In the opinion of the reviewer, the introduction must be revised, thereby focusing on the
Furthermore, no explanation / rationale is given to analyze complement, HMGB1 etc.
Figure legends need formatting.
Cardiomyopathy: Do the authors have clinical data about cardiac function? More details must be provided about cardiac interstitial inflammatory cell infiltration and vasculitis. Also, the authors mention a reduction in CO, but provide no figure / table.
Methods: “As shown in Fig. 6A, anesthetized swine were randomized and subjected to a moderate BI (BOP=350kPa; t+ = 2.5ms)”. It might be helpful for the reader to give concrete examples to visualize the degree of impact (examples of military- and non-military-related incidents that have a comparable impact).
The authors state that “There were no hemodynamically unstable pigs or those with increased lactate levels (> 2mm/L) after surgery”. The definition of shock at least includes a rise in lactat. Please explain.
C3a and HMGB1 - It is not intuitive why the authors have choosen explicitly these markers.
Please provide the rationale / explain the reason for using Plasma-Lyte A.
Reviewer 2 Report
The authors are off to a good start, however, this study requires additional literature review, experiments and explanations. Alternatively, the authors should include more information that clarifies and justifies their choice of methods.
Major concerns:
Results:
General concern: The classical complement upregulation was only measured in blood. It would be good to further evaluate in other tissues via IHC staining of classical complement factors (also other C3s as C3d and C3b, MAC). Further, it would be good to also investigate what the factors are tagging? Neurons ? Astrocytes? Microglia? Is it possible that you see A1 astrocytes? Are the factors intrinsically produced in the brain or is it due to a BBB? Compare to control group.
Which shock tube was used?
Why only measured known organ injury biomarkers? Did you measure any soluble known TBI biomarkers in CSF or blood? GFAP and NF-H are known markers, but what about tau? IL? TNF?NF-L? Please elaborate.
What was your control group and n number?
Figure 1: title not readable
Correlation between increased white blood cells and other pathological abnormalities (increased neuroinflammatory markers, death, increased BBB breakdown)?
Why was only C3a measured? How about C3d and C3b?
Was MAC up regulation investigated? Please discuss hypothesis why only C3a up-regulated? Only opsonisation? Where was this marker expressed? Which cells expressed it? Or which cells were tagged by it? C3 expression uniformly unregulated in whole brain? What was used as control blood or brain tissue?
General concern: Introduction focuses mainly on TBI, however the results start with a very detailed description of lung pathologies, which is indeed important. However, characterization of pathophysiology in brain is insufficient (specific markers are missing and quantification is missing).
Histopathological evaluation is insufficient:
- Why was only H/E satining used to investigate TBI Pathology?
- Add IHC staining of specific TBI markers
- Add IHC of complement and markers
- E.g.; neuronal apoptosis, neuroinflammation, and neurodegeneration please validate with specific markers
- How was correlation of HMGB1/C3 and heart alterations calculated?
- How was the injury score calculated?
- Did I understand it correctly that n=5 samples were stained for H/E and quantified? What was quantified and how? How whee the 30 random images chosen?
What’s the difference between 4.2.1 and 4.2.2.? What were the distinct methods used for? Please refer to results part.
Discussion:
Please elaborate on other models and limitations? In vitro studies (organoids, chips) and small animal studies? Pro and cons.
Please discuss in detail what up regulation of C3a in blood means? Why not other markers? Indicates what? Opsonins only? What are they tagging?
Round 2
Reviewer 1 Report
Thank you for your revision!
Reviewer 2 Report
Accept